# Cannabinoid Type 1 Receptor is Undetectable in Rodent and Primate Cerebral Neural Stem Cells but Participates in Radial Neuronal Migration

**DOI:** 10.3390/ijms21228657

**Published:** 2020-11-17

**Authors:** Yury M. Morozov, Ken Mackie, Pasko Rakic

**Affiliations:** 1Department of Neuroscience, Kavli Institute for Neuroscience, Yale School of Medicine, Yale University, New Haven, CT 6510, USA; 2Gill Center for Biomolecular Science, Indiana University, Bloomington, IN 47405-2204, USA; kmackie@indiana.edu; 3Department of Psychological and Brain Sciences, Indiana University, Bloomington, IN 47405-2204, USA

**Keywords:** CB1 receptor, endocannabinoids, neurogenesis, neuron migration, IHC data interpretation, electron microscopic 3D reconstruction

## Abstract

Cannabinoid type 1 receptor (CB_1_R) is expressed and participates in several aspects of cerebral cortex embryonic development as demonstrated with whole-transcriptome mRNA sequencing and other contemporary methods. However, the cellular location of CB_1_R, which helps to specify molecular mechanisms, remains to be documented. Using three-dimensional (3D) electron microscopic reconstruction, we examined CB_1_R immunolabeling in proliferating neural stem cells (NSCs) and migrating neurons in the embryonic mouse (*Mus musculus*) and rhesus macaque (*Macaca mulatta*) cerebral cortex. We found that the mitotic and postmitotic ventricular and subventricular zone (VZ and SVZ) cells are immunonegative in both species while radially migrating neurons in the intermediate zone (IZ) and cortical plate (CP) contain CB_1_R-positive intracellular vesicles. CB_1_R immunolabeling was more numerous and more extensive in monkeys compared to mice. In CB_1_R-knock out mice, projection neurons in the IZ show migration abnormalities such as an increased number of lateral processes. Thus, in radially migrating neurons CB_1_R provides a molecular substrate for the regulation of cell movement. Undetectable level of CB_1_R in VZ/SVZ cells indicates that previously suggested direct CB_1_R-transmitted regulation of cellular proliferation and fate determination demands rigorous re-examination. More abundant CB_1_R expression in monkey compared to mouse suggests that therapeutic or recreational cannabis use may be more distressing for immature primate neurons than inferred from experiments with rodents.

## 1. Introduction

The cerebral cortex is composed of a complex network of excitatory projection neurons and inhibitory interneurons that are generated, migrate to predetermined destination and establish specific synaptic connections through precisely orchestrated processes, and dysfunction during assembly may cause neuropsychiatric disorders such as lissencephaly, polymicrogyria and autism spectrum disorder [1,2,3,4,5,6,7]. Neural progenitors undergoing repeated cycles of proliferation in the ventricular zone (VZ) undergo interkinetic nuclear translocation whereby the postmitotic nucleus moves within the vertically elongated cell body to the basal region of the VZ during the S phase of the cell cycle and then it returns to the apical region for mitosis [8]. During interkinetic nuclear translocation, cells stay attached to the ventricular surface through adherence junctions between adjacent cells and do not actually migrate but instead, interphase nuclei may be displaced out from the apical region by other dividing and growing cells [9]. After a postmitotic cell detaches from the ventricular surface, it undergoes active migration towards the developing cortical plate where it settles as excitatory projection neuron. A fundamental detail of the molecular mechanism of neuronal nucleus movement and cell body locomotion is coordination of the cytoskeleton reformation by the centrosome—a highly conservative organelle that consists of two centrioles surrounded with matrix proteins and serves as the primary microtubule-organizing center in animal cells [10,11,12]. For example, cells undergoing mitosis cannot actively migrate because their cytoskeletons and particularly centrosomes are engaged in chromosome separation. Interphase neural stem cells (NSCs) in the VZ also do not actively migrate while they have both centrioles included in the structure of primary cilia as basal bodies and daughter centrioles making them unavailable for the cytoskeleton reformation crucial for cell migration. Release of the centrosome from the eliminated primary cilium and its relocation ahead of the migrating nucleus takes place simultaneously with initiation of migration in postmitotic neurons [10,13,14,15,16]. When the migration ends, the centrosome finalizes its function for nuclear movement, attaches to the cell membrane and participates in the recovery of the primary cilium [17,18]. In the present electron microscopic study, we identified the centrosome position in the analyzed cells and used them as a fiduciary organelle for precise identification of the cellular developmental stage.

Retrograde synaptic signaling through CB_1_R is well documented, while many potential functions of the endocannabinoid system are less well understood [19,20,21]. CB_1_R is expressed in the mammalian brain early in development and are involved in vital and complex developmental events such as axonal growth and path finding [22,23,24] and in neuronal migration [25,26]. Endocannabinoid- and CB_1_R-mediated promotion of neural progenitor differentiation into astroglial cellular line has been suggested [27,28,29], however, these developmental events demand more detailed investigation. For example, labeling by anti-CB_1_R sera was observed with light microscopy in neural progenitors and radial glia in vitro and in vivo [30,31,32]; but, precise CB_1_R cellular location, which determines the receptor function, remains to be demonstrated in NSCs. In the present study we analyzed CB_1_R expression and the subcellular location in the proliferating NSCs and postmitotic neurons migrating through transient embryonic zones of the cerebral cortex during the middle neurogenic stage in mice and rhesus macaques. Combining immunohistochemical (IHC) labeling with electron microscopic 3D reconstruction from uninterrupted serial ultrathin sections allowed us to estimate CB_1_R expression and distribution in distinct cells at specific developmental stages.

## 2. Results

In silico analysis of our previous quantitative whole-transcriptome mRNA sequencing data [33] demonstrated varied levels of expression of endocannabinoid system genes in mouse embryo cortical zones. More specifically, we found dramatic upregulation of the CB_1_R gene (*Cnr1*) in parallel with presumed prospective pyramidal neurons relocation from VZ to cortical plate (CP). Low content of CB_1_R in the VZ/subventricular zone (SVZ) raises the question about when CB_1_R is first expressed, which we addressed below. Expression of other genes of the endocannabinoid system was low or absent in all analyzed embryonic cerebral zones with the exception of an increase of *Mgll*—an endocannabinoid degrading enzyme—in samples combining CP and the marginal zone (MZ), while *Mgll* remains very low in VZ/SVZ and intermediate zone (IZ). This may be due either to *Mgll* expression in the interneurons tangentially migrating through the MZ, or its fast upregulation in projection neurons settling in the CP (Figure 1). Low expression of genes encoding endocannabinoid synthetizing enzymes indicates that the ligands of CB_1_R likely are not locally synthetized in the embryonic neocortex. They might be delivered to developing brain through blood circulation from other organs and/or mother organism through placental penetration or breast feeding during postnatal development [34,35].

Next, we performed IHC labeling using different anti-CB_1_R sera followed by several visualization and amplification methods such as immunofluorescence, immunoperoxidase and silver amplification of ultra-small gold particles in the mouse embryo brain. These results indicate that antibodies generated against the last 15 amino acids (L15) of the CB_1_R C-terminus also stain radial processes in addition to the previously demonstrated selective labeling of CB_1_R in cortico-fugal axons and interneuron cell bodies [23,24]. Double immunolabeling for fluorescence and electron microscopy demonstrated colocalization of anti-CB_1_R staining with the radial glia cell marker Glast but not with neuronal markers TuJ1 and doublecortin (Figure 2). Nevertheless, our control tests show equal staining of radial processes in wild type mouse embryos and CB_1_R^−/−^ littermates, identifying this as artifactually positive CB_1_R immunolabeling in radial glia cells (Figure 3). Selective axonal labeling was observed in wild type mice with different CB_1_R antibodies generated against C-terminus or amino-terminus (NH) while it was abolished in CB_1_R^−/−^ embryos or by preincubation of the antibodies with their respective fusion peptides (Figure 4). Out of several antibodies tested in this and our previous studies [24,36,37], CB_1_R antibody generated against last 31 amino acids (L31) provides maximal selectivity with minimal background staining that determined our choice of this antibody for electron microscopic 3D analysis (see below).

In accord with the transcriptome data (Figure 1), our IHC labeling identified distinct amounts of CB_1_R in the embryonic cortical zones (Figure 5). In both the mouse and macaque monkey, VZ and SVZ are mostly CB_1_R-immunonegative whereas bundles of horizontal CB_1_R-positive axons emanating from cortical projection neurons are abundant in the IZ [23]. The MZ contains numerous CB_1_R-positive cell bodies of tangentially migrating interneurons [24]. Mouse CP demonstrated minor CB_1_R accumulation, whereas CB_1_R-positive globules in radially migrating cell bodies (see below for a detailed electron microscopic analysis) are numerous in the macaque monkey (Figure 5).

Our light microscopic observations indicate that VZ/SVZ cells were mostly immunonegative with the exception of sporadic positive vertical processes. The positions of the cell bodies emitting these processes were not visible with light microscopy provoking the hypothesis that they are emitted by VZ radial glia cells. To try to identify these cell bodies, we applied correlative light/electron microscopic analysis and unexpectedly detected CB_1_R-expressing growth cones directed towards the ventricular surface identifying the vertical CB_1_R-positive processes to be unusually oriented growing axons (Figure 6). Most likely, they represent disoriented cortico-fugal axons emitted by developing pyramidal neurons located in the CP [23]. Electron microscopic 3D reconstruction shows that several vertical axons may continuously contact each other revealing reciprocal affinity and contact navigation of growing CB_1_R-expressing axons (Figure 6). Although a detailed study of this phenomenon is beyond the scope of the present article, atypical CB_1_R-expressing axons in the VZ/SVZ indicate a technical opportunity to identify rare CB_1_R-containing structures further confirming immunonegativity of majority of VZ/SVZ cells.

As a next step, we applied electron microscopic 3D reconstruction to analyze cells at distinct developmental stages. A total of 67 randomly chosen cells from wild type mice and monkeys were analyzed; see Appendix A for detailed morphologic characterization of each cell. Mitotic and postmitotic cells in the VZ were either spherical or emitting radial processes (*n* = 9 cells analyzed in this article; see also [38]). Out of a total of 52 interphase cells from SVZ, IZ and CP of wild type mice and monkeys, 50 showed pronounced vertical elongation of the cell body and the nucleus. Out of two non-vertical cells located in the IZ, one was multipolar (cell #M14) with unpronounced orientation of processes [39], and the other one was bipolar and horizontally oriented (cell #M15)—presumed to be a tangentially migrating interneuron, similar to horizontal cells in MZ (*n* = 6 cells). Among the vertical cells from SVZ, IZ and CP, 39 were bipolar; 7 were multipolar with 3 or 4 processes and situated in the basal layer of CP—likely finalizing those vertical migration; 3 were unipolar with basally directed process—likely undergoing vertical somal translocation [40]; and one cell (#R10) in SVZ was unipolar with the apical process. Thus, bipolar cells predominate in developing neocortex, reflecting the massive vertical migration taking place [2].

In accordance with previous publications [10], we observed a characteristic behavior of centrosomes that we used for more substantial identification of the cellular developmental stages. Namely, early prophase cells (*n* = 2) showed centrioles included in the cilia apparatus and duplication of centrosomes located close to the ventricular surface; prometaphase cells (*n* = 2) demonstrated two centrosomes separated for prospective daughter cells and absence of primary cilia. In numerous analyzed VZ and SVZ cells (*n* = 24) in mice and macaque monkeys, we did not find centrosomes in the vicinity of the cell body that indicates they were distantly located in truncated processes; while two cells showed centrosomes in the apical processes indicating that these cells had detached from the ventricular surface and were preparing for basal migration (Figure 7). So, most cells in VZ/SVZ of E16 mice and E45 monkeys are likely in the cycle of interkinetic nuclear translocation or in the stage of initiation of vertical migration. In contrast, in the analyzed vertical cells in IZ and CP (*n* = 36), we identified 30 centrosomes, 26 of which were located basally relative to the nucleus and 4 showed the position aside of the nucleus (Appendix A and Figure 8, Figure 9 and Figure 10). Although centrosomes were missed in some incompletely reconstructed processes, the predomination of basal centrosome position confirms massive basal migration of projection neurons in IZ and CP. Thus, cellular morphology and intracellular location of centrosomes identify interkinetic nuclear translocation of VZ/SVZ cells, initiation of basal migration in SVZ cells and massive locomotion through the IZ and CP.

Despite an extensive search, we did not find credible CB_1_R immunolabeling in VZ or SVZ mitotic and postmitotic cells (*n* = 24 cells occasionally chosen for 3D reconstruction). The trace labeling observed (about one locus of the antibodies binding per 100 μm^3^ of cytoplasm) was similar in wild type and CB_1_R^−/−^ mice indicating that this is the level of non-selective background (Appendix A, Figure 10 and Figure 11). The earliest observation of CB_1_R-L31 immunoreaction end-product depositions is in the cytoplasm of IZ cells. Neurons migrating radially through IZ and CP contained intracellular vesicles surrounded by the immunoreaction end-product forming typical CB_1_R-positive globules previously detected only in the interneurons [36,41]. CB_1_R-positive globules were numerous in nearly all migrating neurons in monkeys, whereas in mice only 4 out of 21 vertical migrating cells contained similar globules that show less intensive labeling. Thus, CB_1_R was undetectable in VZ and SVZ cells while they were undergoing repetitive circles of proliferation or initiating migration, whereas postmitotic neurons migrating radially through IZ and CP express CB_1_R and this expression was more abundant in macaque monkeys than in mice.

Out of 12 migrating neurons randomly chosen for 3D reconstruction from the IZ of CB_1_R^−/−^ mice, eight emitted processes in directions that deviated from basal or apical. One cell had only the apical process, which is also unusual for IZ cells; and only three cells showed a normal bipolar vertical morphology (Appendix A; Figure 11). This contrasts to neurons in the IZ of wild type mice and monkeys where these neurons were mostly bipolar (9 out of 14 cells analyzed; Figure 10). Neurons migrating radially through the CP of CB_1_R^−/−^ mice (*n* = 4) apparently recovered the typical bipolar phenotype (Figure 11). Thus, projection neurons in CB_1_R^−/−^ mice show morphological features of temporarily disoriented migration such as a non-vertically shaped cell body and an increased number of processes emanating in random directions. Delay of correct neuronal migration may contribute to developmental disorders of cerebral cortex and later behavioral consequences for the animal.

## 3. Discussion

Here we applied several methods, such as in silico analysis of whole-transcriptome mRNA data and light and electron microscopic IHC to estimate expression of the endocannabinoid signaling system and particularly CB_1_R in distinct cells at specific developmental stages in mouse and rhesus macaque. Using morphological and ultrastructural criteria such as the laminar position and shape of the cell body and intracellular position of the centrosome, we identified stages of projection neuron generation and migration. We found that CB_1_R in the cell body accumulates in parallel with cell-autonomous development, namely: mitotic and postmitotic cells in VZ and SVZ were immunonegative, whereas the cells migrating through IZ and CP contain CB_1_R-positive intracellular vesicles. Visible CB_1_R immunoreactivity of vertical processes emanating from radial glia cells may be non-selective due to nonspecific immunolabeling conditions. Simultaneously, relatively rare authentic CB_1_R-positive vertical processes in the VZ/SVZ may represent disoriented axons most likely emanating from cortical projection neurons [23]. We conclude that the low levels or absence of CB_1_R from proliferating VZ/SVZ cells indicates that previously suggested direct influence of the cannabinoid signaling system and particularly CB_1_R to cell proliferation and developmental fate of NSCs [29] should be rigorously examined. Observed actions of cannabis and endocannabinoids on proliferating neurons [27,28] may be less direct and, for example, may be transmitted through other molecules, cells, organs or the maternal organism. Potential actions of CB_1_R in later prenatal and postnatal proliferation zones such as the neocortical outer SVZ or hippocampal dentate gyrus [42] demands further investigation using methods with high cellular and subcellular resolution.

Consistent with previous studies of developing mice [25,26], here we showed that initial expression of the CB_1_R gene and accumulation of the protein in the cell body coincided with the initiation of postmitotic neurons migration towards their final positions in the developing cortical plate. Block of endocannabinoid signaling in CB_1_R^−/−^ mice causes a deviation from the correct vertical migration that is detectable as a non-vertical shape of the cell bodies, which also emit numerous disoriented processes. Such a cell migration disorder may be temporary and its overall impact only moderate for the animal’s ontogenesis consistent with the relatively normal cortical development of CB_1_R^−/−^ mice [43]. Nevertheless, as we show here, CB_1_R expression in cerebral projection neurons is more pronounced in monkeys than in the mouse as judged by the number of CB_1_R-positive cells and quantity and intensity of labeling of CB_1_R-containing intracellular vesicles in each cell. Therefore, medicinal or recreational cannabis use may affect immature primate neurons more than is apparent from rodent experiments. CB_1_R-containing vesicles may play a role in intracellular self-regulation sharing pathways similar to those used in slow self-inhibition, which is elicited by cell autonomous Ca^2+^-dependent production of endocannabinoids and K^+^ channel activation, as previously detected in cholecystokinin/CB_1_R expressing interneurons and a subpopulation of cortical pyramidal neurons in adult mice [44,45]. Fluctuations of Ca^2+^ concentration in the somata may influence neuronal migration, likely through the reformation of a microtubule and other components of the cytoskeleton [46,47,48]. Thus, CB_1_R-containing intracellular vesicles in migrating neurons represent a potential molecular substrate for endocannabinoid participation and cannabis disruption of cytoskeleton reorganization through regulation of intracellular Ca^2+^ oscillations.

## 4. Materials and Methods

### 4.1. In Silico Quantification of Cannabinoid System Genes Expression in the Mouse Embryo Cerebral Cortex

Quantitative whole-transcriptome mRNA sequencing data were obtained from Rakic lab website https://medicine.yale.edu/lab/rakic/transcriptome/ (accessed on 10 July 2013). The data are also publicly available at the online base of National Center for Biotechnology Information at https://www.ncbi.nlm.nih.gov.

### 4.2. Animals

Animal protocols for non-human primates and mice were complied with the NIH guidelines for animal care and use and were approved by the Institutional Animal Care and Use Committee of Yale University (Protocol #2018-10297.A2 approved 22 July 2020; protocol #2018-10750.A3 approved 25 June 2020). Timed-pregnant macaque and mice were housed in the vivarium of Yale University on a recommended diet and socializing. For terminal surgery, the animals were anaesthetized with sodium pentobarbital (3 mL/kg) or euthasol (0.5 mL/kg). Two monkey embryos at E45 were collected through cesarean sections performed by veterinarians. Fetal age was estimated based on weeks after ovulation, crown-rump length and anatomical landmarks. No apparent abnormalities were noted at the time of tissue collection. Monkey embryos were decapitated, brains were removed and one hemisphere was used for ex vivo analyses for other projects while the other was immersed in fixative containing 4% paraformaldehyde, 0.2% picric acid and 0.2% glutaraldehyde in 0.1 M phosphate buffer (pH 7.4) for 3–4 days at 4 °C. Timed-pregnant mice were killed with cervical translocation and the embryos exteriorized from the uterus and decapitated, and their brains removed and fixed as above. Wild type C57BL6 mouse embryos at E14 (*n* = 4), E15 (*n* = 8) and E16 (*n* = 4) and CB_1_R-knock out (CB_1_R^−/−^) embryos in C57BL6 background (generated at NIMH, Bethesda, MD [43]) at E14 (*n* = 4) and E15 (*n* = 3) were used for morphologic analyses.

### 4.3. IHC Labeling for Light and Electron Microscopy

IHC labeling was performed as previously described [37,49,50]. Briefly, coronal 80-m-thick slices from the embryo cerebral cortex were cut by a vibratome. For double immunofluorescence labeling, slices were blocked in solution of 5% bovine albumin with 0.05% Triton X-100 and immersed in the mixture of rabbit polyclonal antibodies generated against the last 15 amino acids of rat CB_1_R (CB_1_R-L15; dilution 1:1000 [51]), with guinea pig Glast polyclonal antibodies (Synaptic Systems, Goettingen, Germany; catalog number 250114; 1:5000), monoclonal mouse neuronal class III-tubulin antibodies (TuJ1; Covance, Berkeley, CA; catalog number MMS-435P; 1:1000), or goat doublecortin (C-18) polyclonal antibodies (Santa Cruz Biotechnology, Dallas, TX; catalog number sc-8066; 1:1000) overnight at room temperature. Then slices were immersed in a mixture of the corresponding combination of secondary antibodies conjugated with Alexa A-594 and Alexa A-488 (all from Molecular Probes, Eugene, OR; 1:300) and mounted on microscope slides. Slices were evaluated and photographed in microscope Axioplan 2 equipped with the Apotom system (Zeiss, Jena, Germany). For electron microscopy, vibratome slices were cryoprotected in 30% sucrose, freeze–thawed over liquid nitrogen, blocked in 5% bovine albumin and incubated in rabbit CB_1_R-L15 (1:1000), rabbit polyclonal antibodies generated against amino-terminus of rat CB_1_R (CB_1_R-NH; 1:2000 [52]) or guinea pig polyclonal antibodies generated against last 31 amino acids of mouse CB_1_R (CB_1_R-L31; Frontier Science Co, Ltd., Japan; catalog number CB1-GP-Af530-1; 1:2000) overnight at room temperature. Then slices were immersed in respective solutions of biotinylated secondary antibodies (Jackson Immunoresearch Inc., West Grove, PA; 1:300) and developed by the Elite ABC kit (Vector Laboratories, Burlingame, CA) with Ni-intensified 3,3′-diaminobenzidine-4HCl (DAB-Ni) as a chromogen. For double immunolabeling, we first applied CB_1_R-L15 serum as above and goat antirabbit IgGs conjugated with 1 nm gold particles (1:80) with the subsequent silver intensification kit R-Gent SE-LM (all from Aurion, Wageningen, The Netherlands). Then, Glast was visualized with made-in-guinea pig antibodies and the immunoperoxidase reaction as above with 3,3′-diaminobenzidine-4HCl (DAB) as a chromogen. Specificity of CB_1_R antibodies was tested in CB_1_R^−/−^ embryos and by preincubation of the primary antibodies (dilution as above) with corresponding fusion proteins (concentration 1 g/mL) for 2 h before the immunoreaction. See also our previous articles [37,53] for control tests in CB_1_R^−/−^ animals. Slices were post-fixed with 1% OsO_4_, dehydrated and embedded in Durcupan (ACM; Fluka, Buchs, Switzerland) on microscope slides. Selected neocortex segments were photographed with an Axioplan 2 microscope (Zeiss, Jena, Germany).

### 4.4. Serial Ultrathin Sectioning and Electron Microscopic 3D Reconstruction

For electron microscopic investigations, fragments from neocortical zones were re-embedded into Durcupan blocks and cut by a Reichert ultramicrotome into 70-nm-thick sections. Series of 150–200 consecutive sections were collected on one-slot grids covered with Butvar B-98 films (EMS, Hatfield, PA, USA), stained with lead citrate and evaluated in the JEM 1010 electron microscope (JEOL, Tokyo, Japan) equipped with the Multiscan 792 digital camera (Gatan, Pleasanton, CA, USA) or Talos L120C electron microscope (ThermoFisher Scientific, Boston, MA, USA). Serial images of mitotic or interphase cells from identified zones of the embryo neocortex were made with 12,000× or 15,000× magnification of JEM1010 or with 3400× of the Talos L120C electron microscope. 3D reconstruction of cell bodies with proximal segments of the processes and measurement of the volume and volume of the nucleus or chromosomes were performed using the computer program Reconstruct 1.1.0.0. (Boston, MA) [54,55] publicly available at http://www.bu.edu/neural/Reconstruct.html.

Mitotic phases of the 3D reconstructed cells were identified by the intracellular position of duplicated centrosomes and condensed chromosomes. Projecting processes were identified by their continuity with the cell membrane and a diameter of at least 0.5 μm in contrast to filopodia of 0.1–0.2 μm diameter. Growth cones were identified through the termination of the process with numerous filopodia [23]. The migration stage of interphase cells was determined through the laminar position and shape of the cell body with projecting processes and the intracellular location of the centrosome (if identified) relative to the nucleus. Centrosome position relative to the nucleus was defined as basal, apical or aside of the nucleus. The following morpho-functional/developmental stages of centrosomes were identified in the interphase cells: (1) both centrioles are in the cytoplasm; (2) the mother centriole is attached to the cilial vesicle in the cytoplasm; (3) the mother centriole is attached to the cell membrane, no cilia is formed and (4) both centrioles are included in the cilia apparatus as the basal body and daughter centriole [17]. Following migration, stages of interphase cells were identified: (1) interkinetic nuclear translocation—VZ/SVZ cells with adherent junctions and cilium on the ventricular surface; (2) initial vertical migration—bipolar cells in VZ/SVZ detached from the adherent junctions with the centrosome in the apical segment; (3) locomotion—bipolar vertical cells in IZ/CP with the centrosome located basally or aside of the nucleus; (4) terminal vertical migration—multipolar vertical cells in CP; (5) multipolar migration—multipolar cells in IZ [39]; (6) somal translocation—radially migrating cells in IZ/CP with the single basal process [40] and (7) lateral migration—horizontal bipolar cells in IZ or MZ. We considered interkinetic nuclear translocation and initial radial migration stages of VZ/SVZ cells indistinguishable from each other if centrosomes were not detected in the serial sections. Anti-CB_1_R DAB-Ni depositions in each cell were traced for 3D reconstruction and counted as “single” depositions if local staining were identified in at least three serial sections; merging conglomerates of the immunoreaction end-product around intracellular vesicles were counted as “globule” depositions. Numbers of “single” and “globule” depositions from each cell were normalized per 100 μm^3^ of the analyzed cytoplasm and average ± SEM was calculated for each embryonic cortical zone in wild type animals or totally for CB_1_R^−/−^ embryos. Although such quantification obviously underestimates immunolabeling in the cells with a high level of CB_1_R expression due to merging of adjacent DAB-Ni depositions, it is conclusive for low labeling.

## 5. Conclusions

1. Low, if any, level of CB_1_R in proliferating cells indicates that direct CB_1_R-transmitted regulation of cellular proliferation and cell fate determination is unlikely.

2. CB_1_R in migrating neurons provides the molecular substrate for regulation of cell movement. Block of endocannabinoid signaling in CB_1_R^−/−^ mice provokes migration abnormalities in projection neurons.

3. More abundant CB_1_R expression in the monkey compared to the mouse suggests that therapeutic or recreational cannabis use may be more distressing for immature human neurons than inferred from experiments with rodents.

## Figures and Tables

**Figure 1 ijms-21-08657-f001:**
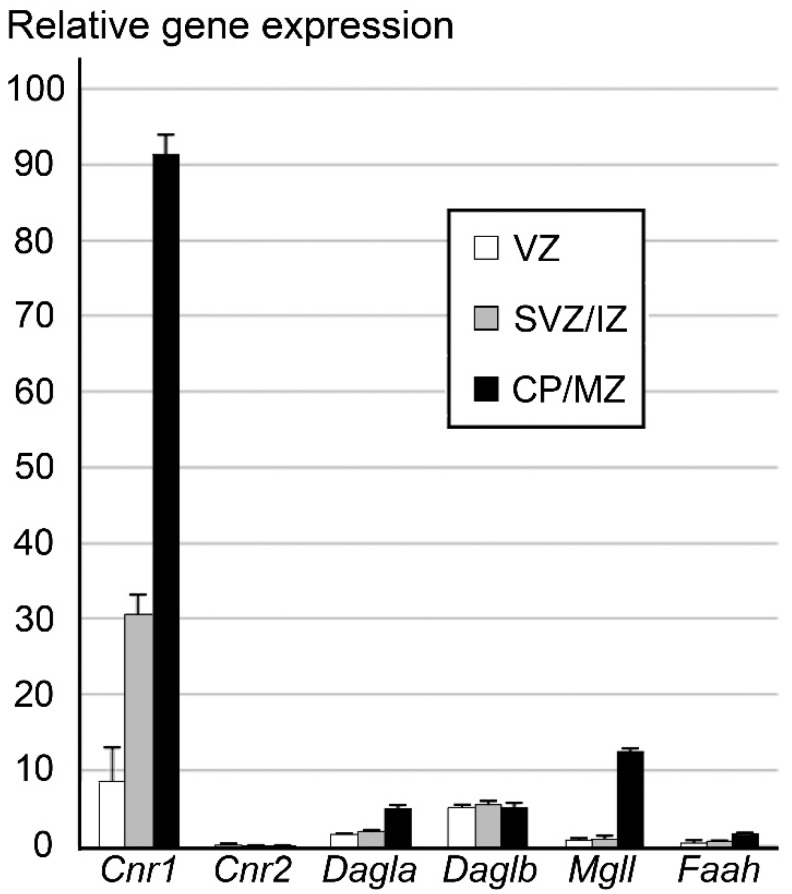
Quantitative whole-transcriptome mRNA sequencing data of the endocannabinoid system genes in the cortical zones from the embryonic day (E) 14.5 mouse embryo. *X* axis shows endocannabinoid system genes: *Cnr1* and *Cnr2* cannabinoid type 1 and type 2 receptors, respectively; endocannabinoid synthesizing enzymes *Dagla* and *Daglb*, sn-1-specific diacylglycerol lipase alpha and beta, respectively; endocannabinoid metabolizing enzymes *Mgll* and *Faah*, monoacylglycerol lipase and fatty acid amide hydrolase, respectively. *Y* axis shows the abundance of the genes by calculating reads per kilobase of the exon model per million mapped reads; average ± SEM from 3 embryos analyzed [33]. Notice dramatic upregulation of CB_1_R gene expression in parallel with neuronal maturation and relocation from VZ to CP. Expression of other genes of the endocannabinoid system is low if any in all analyzed embryonic cerebral zones making doubt for the capacity of immature cells to synthesize endocannabinoids. Increase of endocannabinoid degrading enzyme *Mgll* in CP/MZ may be due to its expression in the interneurons tangentially migrating through MZ or projection neurons settling in CP.

**Figure 2 ijms-21-08657-f002:**
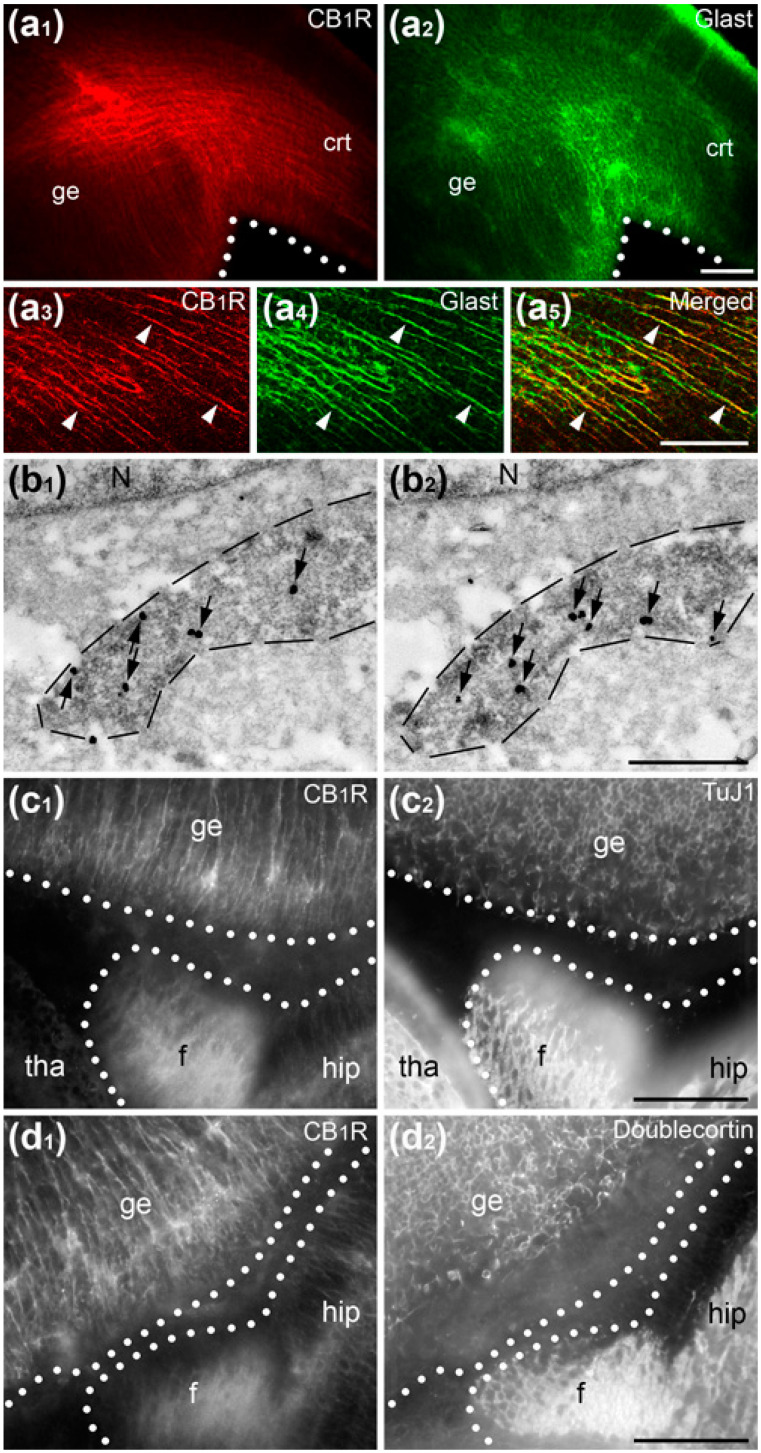
Suboptimal conditions of immunolabeling results in the illusion of CB_1_R expression in E15 mouse embryo radial glia. (**a_1_**–**a_5_**) Colocalization of CB_1_R-L15 (red) and Glast (green) immunolabeling in radial processes in the ganglionic eminence (ge) and the cerebral cortex (crt). (**a_3_**–**a_5_**) High power immunofluorescence images of double labeled radial processes (arrowheads) in the ganglionic eminence. (**b_1_**–**b_2_**) Serial electron micrographs of a radial process (underlined with dashed lines) double-labeled for CB_1_R-L15 (immunogold/silver particles, arrows) and Glast (diffuse DAB immunoperoxidase labeling) in the thalamus. (**c_1_**,**c_2_**,**d_1_**,**d_2_**) Double immunofluorescence reveals dissimilar patterns for CB_1_R-L15 labeling versus neuronal markers TuJ1 (**c_1_, c_2_**) and doublecortin (**d_1_**,**d_2_**) in the ganglionic eminence (ge), thalamus (tha) and hippocampus (hip). Ventricular surfaces are underlined with white dotted lines. Abbreviations: f, fimbria; N, cell nucleus. Scale bars: (**a**,**c**,**d**) 100 μm and (**b**) 1 μm.

**Figure 3 ijms-21-08657-f003:**
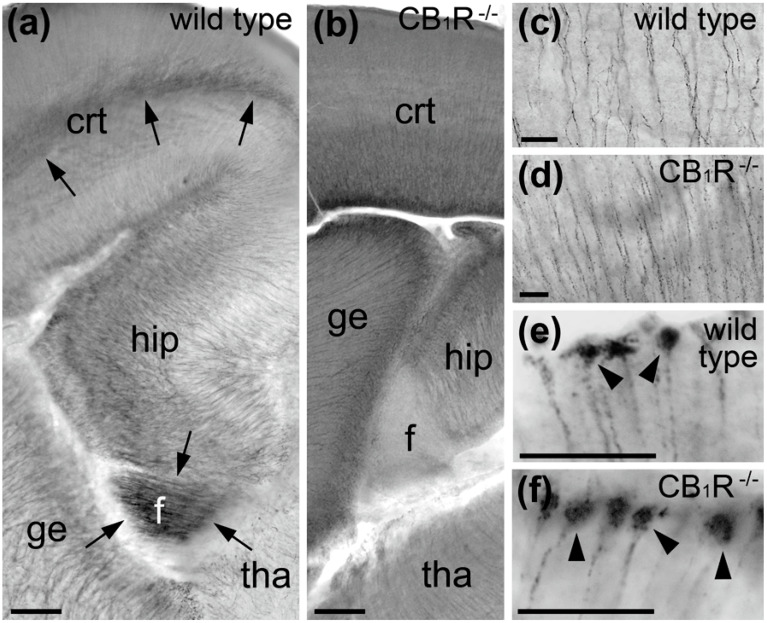
Immunoperoxidase DAB-Ni labeling with CB_1_R-L15 serum in E15 mouse telencephalon. (**a**,**b**) Only axon bundles (arrows) in IZ of the cerebral cortex (crt) and fimbria (**f**) represent selective CB_1_R labeling in wild type embryo as they are immunonegative in the CB_1_R^−/−^ embryo at the same labeling conditions. Radial processes in the dorsal thalamus (tha), ganglionic eminences (ge), cerebral cortex and hippocampus (hip) show non-selective labeling that is similar in wild type and CB_1_R^−/−^ embryos as further confirmed with high power light microscopy images from middle segments of the dorsal thalamus (**c**,**d**) and radial glia cell end-feet (arrowheads) on the pial surface of the lateral ganglionic eminence (**e**,**f**). Scale bars: (**a**,**b**) 100 μm and (**c**–**f**) 20 μm.

**Figure 4 ijms-21-08657-f004:**
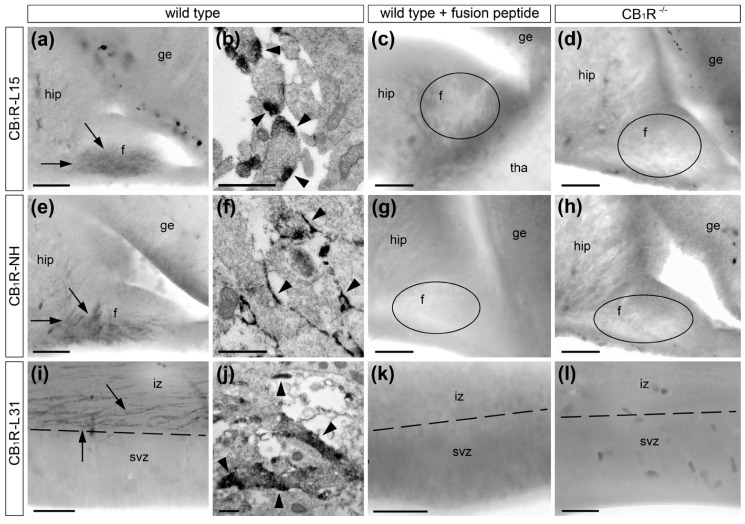
CB_1_R immunolabeling in E15 mouse hippocampal fimbria with CB_1_R-L15 (**a**–**d**) and CB_1_R-NH (**e**–**h**) antibodies and in E14 neocortex with CB_1_R-L31 (**i**–**l**) antibodies. Selective axonal labeling (arrows) in the fimbria (**f**) and neocortical intermediate zone (iz) was observed in wild type embryos with light (**a**,**e**,**i**) and electron microscopy (**b**,**f**,**j**). In accord with the location of CB_1_R C-terminus on the inner surface of the cell membrane and NH-terminus on the outer surface, the DAB-Ni immunoperoxidase reaction end-product (arrowheads) was located inside or outside of the axons, correspondingly. Immunolabeling was abolished by preincubation of the antibodies with the corresponding fusion peptides (**c**,**g**,**k**) and in CB_1_R^−/−^ littermate embryos (**d**,**h**,**l**). Unlabeled fimbria in (**c**,**d**,**g**,**h**) are indicated with ovoids. Border between subventricular zone (SVZ) and intermediate zone is indicated with a dashed line. Abbreviations: ge, ganglionic eminence; hip, hippocampus; tha, thalamus. Scale bars: (**a**,**c**–**e**,**g**–**i**,**k**,**l**) 50 μm and (**b**,**f**,**j**) 0.5 μm.

**Figure 5 ijms-21-08657-f005:**
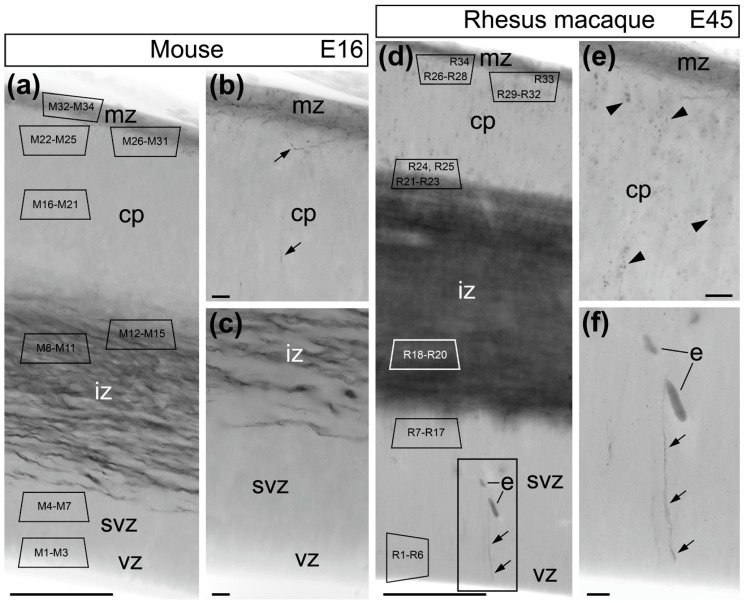
CB_1_R-L31 immunolabeling in the cerebral cortex of E16 mouse (**a**–**c**) and E45 rhesus macaque embryos (**d**–**f**). Abundant immunoperoxidase reaction end-product in the intermediate zones (IZs) of the mouse and monkey are mostly due to CB_1_R-positive axons, whereas marginal zones (MZs) contain numerous CB_1_R-positive cell bodies of tangentially migrating interneurons [23,24]. (**b**) Mouse cortical plate (CP) does not demonstrate CB_1_R accumulation with exception of rare CB_1_R-positive processes (arrows). (**e**) Numerous CB_1_R-positive globules are detectable in the monkey cortical plate (arrowheads). (**c**,**f**) Ventricular (VZ) and subventricular zones (SVZs) are mostly CB_1_R-immunonegative in both animals with exception of rare radial processes (arrows). Area framed with rectangle in (**d**) is enlarged in (**f**). Indexes M1-M34 and R1-R34 inside trapezoid frames in (**a**) and (**d**) schematically indicate positions of 3D reconstructed cells (see below). Abbreviation: e, erythrocytes in blood capillary. Scale bars: (**a**,**d**) 100 μm m and (**b**,**c**,**e**,**f**) 10 μm.

**Figure 6 ijms-21-08657-f006:**
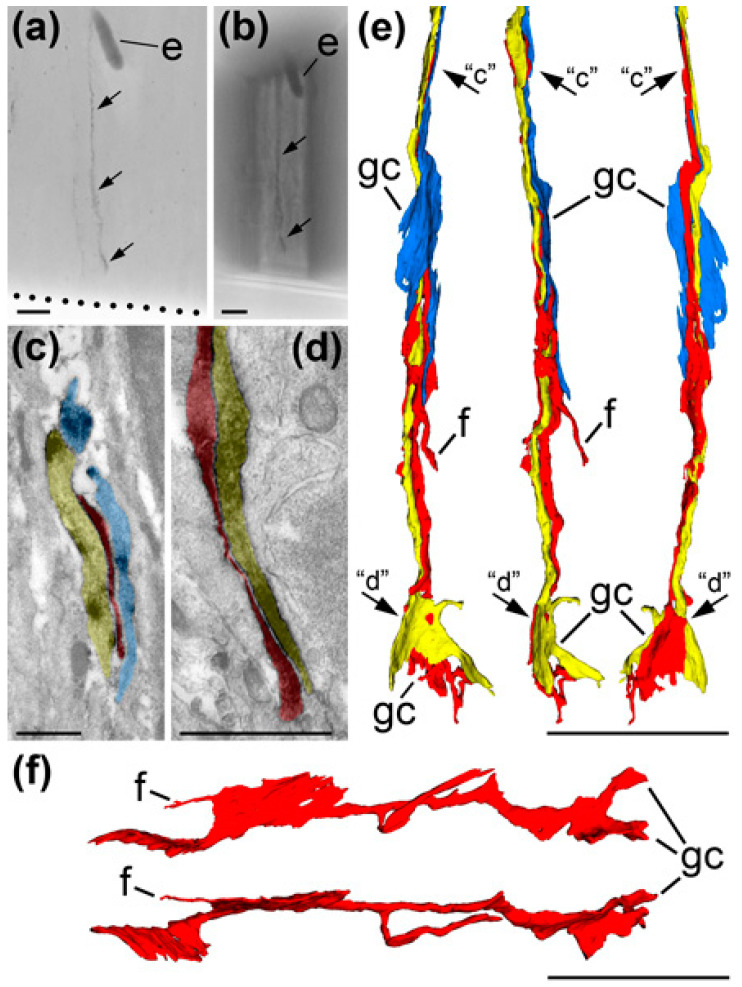
Correlative light-electron microscopic analysis and 3D reconstruction identify CB_1_R-positive radial processes in the cortical VZ of E45 monkey embryo as growing axons directed towards the ventricular surface and ending with growth cones (gcs). (**a**) Light microscopic image (repeated from Figure 5f) of a vertical CB_1_R-positive process (arrows) in VZ. Ventricular surface is indicated with the dotted line. (**b**) Same tissue segment trimmed for serial ultrathin sectioning. Notice trapezoid edges of the sample and the CB_1_R-positive process (arrows). Systematic analysis of serial electron microscopic images (exemplified in (**c**,**d**)) resulting in 3D reconstruction (**e**) of the CB_1_R-positive processes identifies that it represents three axons (correspondingly depicted semitransparent red, yellow and blue in electron micrographs and same colors in 3D reconstruction) tightly following each other. Three 3D images in (**e**) are reciprocally rotated 90 degrees. Arrows “c” and “d” indicate approximate positions of the CB_1_R-positive axon segments shown in (**c**,**d**), respectively. (**f**) Another example of the CB_1_R-positive growth cone in VZ. Two 3D images are reciprocally rotated 90 degrees and placed with growth cone directed right while in the tissue it was directed towards ventricular surface. Notice that all growth cones (gcs) are flattened and end with sharp tips. While vast majority of CB_1_R-positive axons at this developmental stage are growing horizontally through IZ (see Figure 5), vertical orientation of these axons and position of the growth cones close to ventricular surface identify those misorientation. Abbreviation: e, erythrocyte in blood capillary; f, filopodium. Scale bars: (**a**,**b**,**e**,**f**) 10 μm and (**c**,**d**) 1 μm.

**Figure 7 ijms-21-08657-f007:**
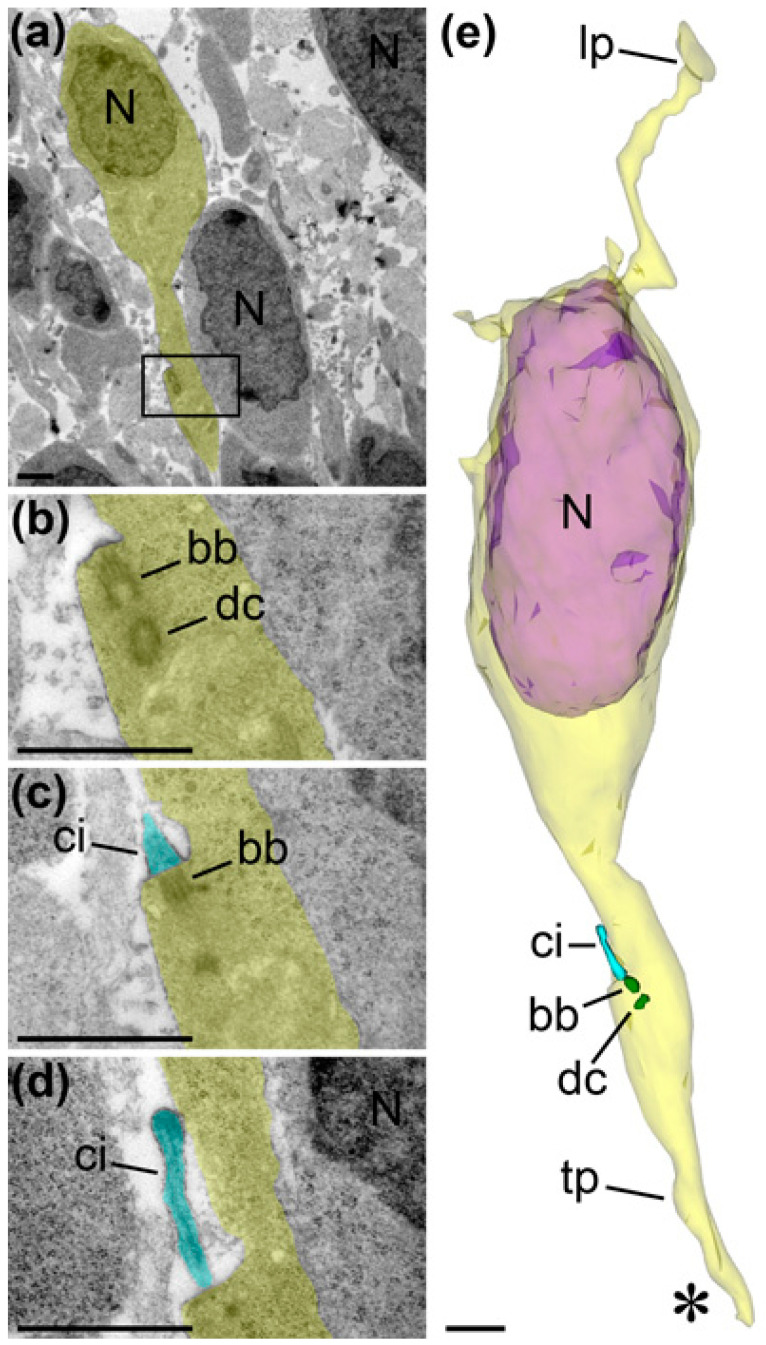
Selected serial electron microscopy images and 3D reconstruction of a postmitotic cell (also shown in Figure 10 as cell R8) from SVZ of E45 rhesus macaque. The images are placed with pia-directed segment up and the segment facing ventricular surface down. Cell body and processes are shown as semitransparent yellow, and the nucleus is semitransparent violet. Framed area in (**a**) is enlarged in (**b**) and further followed in serial images (**c**,**d**). Cilium (ci; depicted light blue) in the apical process designates that the cell is disconnected from the ventricular surface and can initiate vertical migration. Basal body (bb) and daughter centriole (dc) are depicted green in 3D reconstruction. Terminated end of the apical process is indicated with asterisk. Abbreviations: lp, leading process; N, cell nucleus; tp, trailing process. Scale bars: 1 μm.

**Figure 8 ijms-21-08657-f008:**
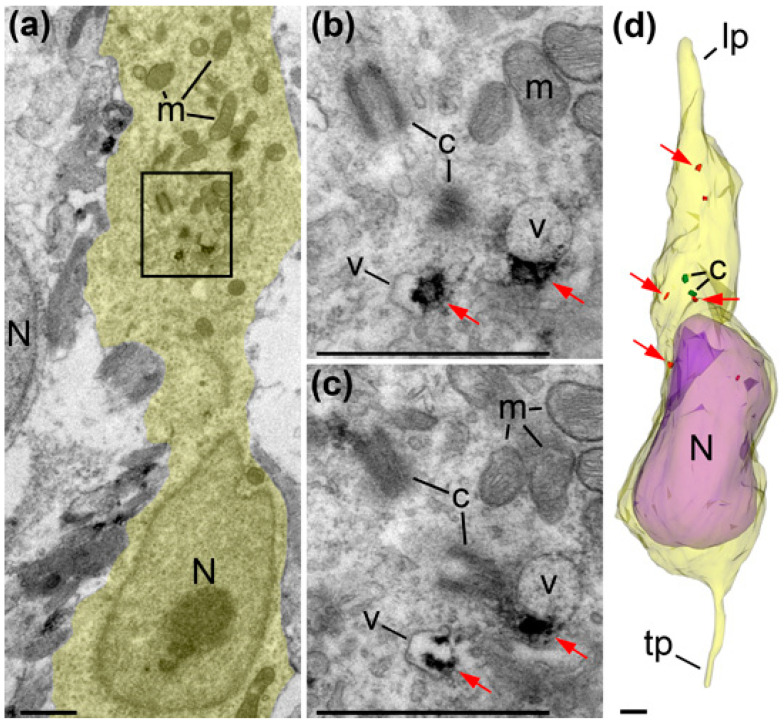
Electron micrographs and 3D reconstruction of a cell radially migrating through IZ of E16 mouse (also shown in Figure 10 as cell M11). The images are placed with the pia-directed segment up and the segment facing ventricular surface down. Cell body and proximal segments of truncated leading (lp) and trailing processes (tp) are shown as semitransparent yellow, and the nucleus (N) is semitransparent violet. Framed area in (**a**) is enlarged in (**b**) and further followed in serial image (**c**). Notice both centrioles (c; depicted green in (**d**)) located basally relative to the nucleus (N). Anti-CB_1_R-L31 immunoreaction DAB-Ni end-products are shown red and pointed with red arrows. Although CB_1_R is not abundant in this cell, its characteristic location on the outer surface of intracellular vesicles (v) proves selective immunolabeling. Abbreviation: m, mitochondria. Scale bars: 1 μm.

**Figure 9 ijms-21-08657-f009:**
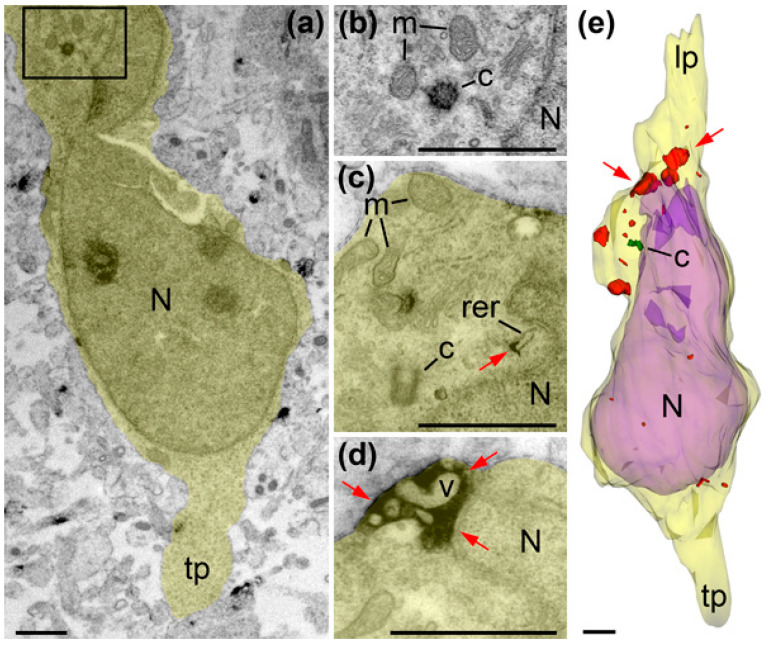
Electron micrographs (**a**–**d**) and 3D reconstruction (**e**) of a vertical bipolar cell in IZ of E45 rhesus macaque (also shown in Figure 10 as cell R23). The images are placed with a pia-directed segment up and the segment facing ventricular surface down. Cell body and proximal segments of truncated leading (lp) and trailing processes (tp) are shown semitransparent yellow, and nucleus (N) is semitransparent violet. Framed area in (**a**) is enlarged in (**b**). Serial not adjacent images (**b**,**c**) show both centrioles (c; depicted green in (**e**)). Small anti-CB_1_R-L31 immunoreaction deposition (red arrow in **c**) on the surface of cisternae of rough endoplasmic reticulum (rer) likely identifies a site of CB_1_R biosynthesis. Numerous intracellular vesicles (v) are surrounded by abundant anti-CB_1_R depositions (red arrows in (**d**)) forming big CB_1_R-positive globules such as shown in red in (**e**). Abbreviation: m, mitochondria. Scale bars: 1 μm.

**Figure 10 ijms-21-08657-f010:**
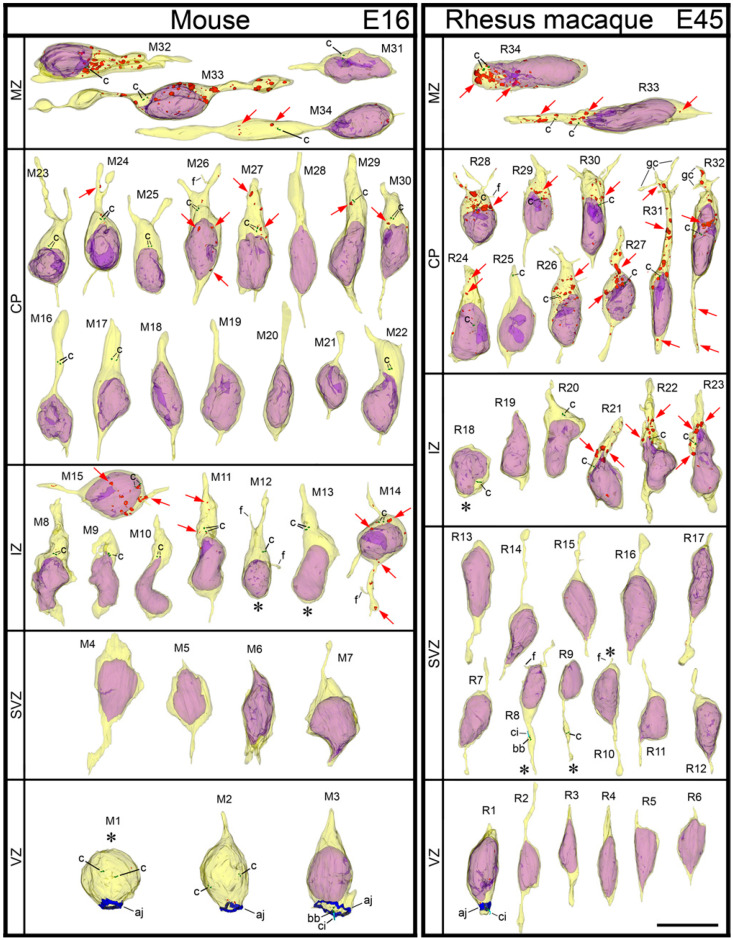
Montage of 3D reconstructed cells from indicated neocortical zones of E16 mouse and E45 rhesus macaque embryos. The cells are placed with pia-directed segments up and the segments facing ventricular surface down. Reconstructed cell bodies with proximal segments of processes are shown as semitransparent yellow, and nuclei are semitransparent violet. Nuclei are absent from mitotic cells while chromosomes in them are not shown. Most of the identified processes are incompletely reconstructed because of truncation in the serial sections. Short processes or cell body segments indicated with asterisks are terminated in the analyzed serial sections. Belts of adherens junctions (aj) connecting the cells at the ventricular surface are shown in blue. Bipolar shape of the cells and centrioles (c; depicted green) located in the leading processes designate radial migration of the cells. CB_1_R-L31 immunoreaction end-product depositions are shown red and some of them also pointed with red arrows. Notice gradual upregulation of CB_1_R while the cells progressively relocate towards CP. Most of the interneurons tangentially migrating through MZ show high CB_1_R content. Primary cilia (ci) are shown as light blue. Indexes of the cells correspond to Appendix A that qualitatively and quantitatively characterizes each cell. The scale bar (10 μm) is valid for all the cells. Abbreviations: bb, basal body; dc, daughter centriole; f, filopodia; gc, growth cone.

**Figure 11 ijms-21-08657-f011:**
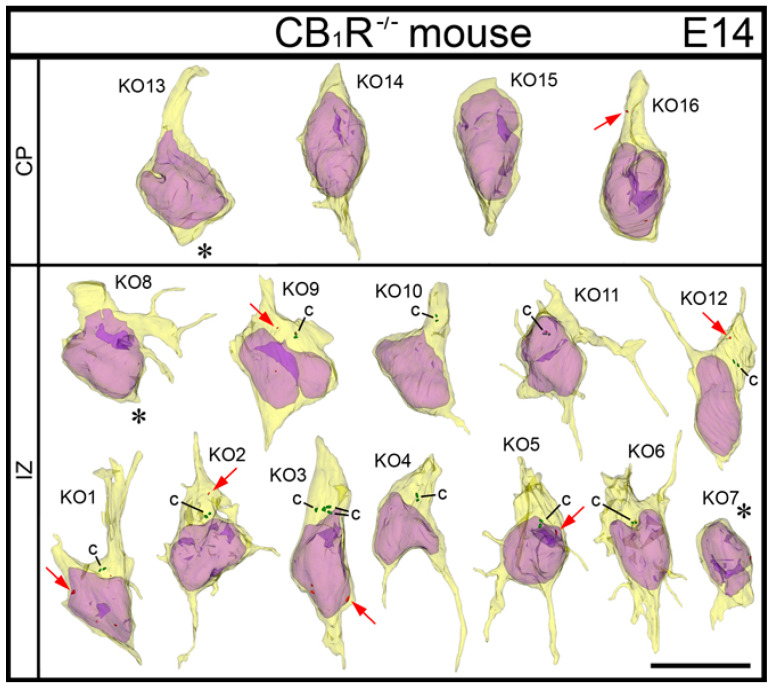
Montage of 3D reconstructed cells from indicated neocortical zones of E14 CB_1_R^−/−^ mouse embryos. The cells are placed with pia-directed segments up and the segments facing the ventricular surface down. Reconstructed cell bodies with proximal segments of processes are shown as semitransparent yellow, and nuclei are semitransparent violet. Most of the identified processes are incompletely reconstructed because of truncation in the serial sections. Short processes or cell body segments indicated with asterisks are terminated in the analyzed serial sections. Most of the cells show multipolar shape designating deviations from normal radial migration. Rare non-selective anti-CB_1_R-L31 immunoreaction end-product depositions are shown as red and pointed with red arrows. Indexes of the cells correspond to Appendix A that qualitatively and quantitatively characterizes each cell. Cell KO3 contains four centrioles (c; depicted green) that identify duplication of the centrosome during early prophase of mitosis. Scale bar (10 μm) is valid for all the cells.

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
