# Peer review of "Cannabinoid Type 1 Receptor is Undetectable in Rodent and Primate Cerebral Neural Stem Cells but Participates in Radial Neuronal Migration"

_ijms, 2020, doi:10.3390/ijms21228657_

Round 1

Reviewer 1 Report

The current manuscript by Morozov et al investigated the cellular location of CB1 receptor in the embryonic mouse and rhesus macaque to clarify the role of endocannabinoid systems in cerebral cortex embryonic development. Moreover, the authors showed that projection neurons in the IZ showed migration abnormalities in CB1 receptor KO mice. Thus, this study has provided beneficial results to understand the function of CB1 receptor in developmental stage. However, there are some critical issues that should be addressed.

Major Issues:

  1. The authors used CB1R-L31 antibody (Frontier Science Co., Catalog number CB1-GP-Af530-1) for tissue staining in the current manuscript. Although they explained “After numerous pilot study, we determined that the optimal CB1R immunolabeling of embryo brain is achieved with CB1R-L31 antibodies and …..” (page 5, line 216-218), they haven't revealed the details. This Ab information is very important for their manuscript. They need to show at least the data following 1) the antibody did not cause nonspecific staining in the samples from CB1R KO mice and 2) the antigen peptide can block the staining of samples from rhesus macaque.

  1. They should add the Y-axis title in Fig1.

Author Response

Reviewer 1.

The current manuscript by Morozov et al investigated the cellular location of CB1 receptor in the embryonic mouse and rhesus macaque to clarify the role of endocannabinoid systems in cerebral cortex embryonic development. Moreover, the authors showed that projection neurons in the IZ showed migration abnormalities in CB1 receptor KO mice. Thus, this study has provided beneficial results to understand the function of CB1 receptor in developmental stage. However, there are some critical issues that should be addressed.

Major Issues:

  1. The authors used CB1R-L31 antibody (Frontier Science Co., Catalog number CB1-GP-Af530-1) for tissue staining in the current manuscript. Although they explained “After numerous pilot study, we determined that the optimal CB1R immunolabeling of embryo brain is achieved with CB1R-L31 antibodies and …..” (page 5, line 216-218), they haven't revealed the details. This Ab information is very important for their manuscript. They need to show at least the data following 1) the antibody did not cause nonspecific staining in the samples from CB1R KO mice and 2) the antigen peptide can block the staining of samples from rhesus macaque.

 Answer:

We agree that mentioning of not shown pilot data may be confusing. We rephrased the sentence on page 5 (lines 139-142) as follow:

Out of several antibodies tested in this and our previous studies (Morozov and Freund, 2003; Morozov et al., 2009; Morozov et al., 2016), CB1R antibody generated against last 31 amino acids (L31) provides maximal selectivity with minimal background staining that determined our choice of this antibody for electron microscopic 3D analysis (see below).

We also included in Figure 4 new panels showing absence of CB1R-L31 labeling after antigen peptide pre-absorption and in CB1R-KO mice.

Patterns of CB1-L31 immunolabeling in mouse and monkey are generally identical demonstrating only quantitative difference. This makes negative control in monkey to be redundant that we, unfortunately, should cancel because of restrictions for unnecessary laboratory activity during COVID-19 pandemic conditions.

  1. They should add the Y-axis title in Fig1.

Answer:

We included following title for Y axis: “Relative gene expression”

Reviewer 2 Report

The study is very well conducted and results convincing. I have only minor comments:

  • will it be possible to show low magnification images for figure 2, 'a' panels.
  • also it will be good to show DCX and TuJ1 stainings, maybe as supplemental?
  • my main concern is regarding the use and labelling of the CB1R antibodies throughout the text/figures as there are some things not completely clear to me. Specifically, it is not clear which antibody was used in figure 4 upper panels...was the L15 or L31? The one written is L15, and it seems specific, but in the previous figure was claimed as aspecific in some regions and in the results section it is written that for the rest of the study (figure 4 or 5 onwards?) it was used a different one (L31). It is also not clear to me why the NH one was discarded or of NH Ab and L31 are the same... Could the authors check these points and help the reader to correctly interpret this section of results and corresponding figures?

Author Response

Reviewer 2.

The study is very well conducted and results convincing. I have only minor comments:

  • will it be possible to show low magnification images for figure 2, 'a' panels.

Answer:

We included low power images for double immunofluorescence that show similar patterns of CB1R and Glast labeling in mouse embryo. We also enlarged fragments from former panels ‘a1-a3’ that show colocalization of CB1R and Glast in distinct radial processes.

  • also it will be good to show DCX and TuJ1 stainings, maybe as supplemental?

Answer:

Probably, reviewer means “DCX and TuJ1 staining in monkey” because we did show it in mouse.

Although DCX and TuJ1 labeling may be helpful for identification of immature neurons, monkey (and mouse) cells at studied developmental stage are unequivocally identified with laminar location in the embryo cerebrum and the 3D ultrastructure. We prefer avoiding optional experiments that are restricted during current COVID-19 conditions. Thank you for understanding.

  • my main concern is regarding the use and labelling of the CB1R antibodies throughout the text/figures as there are some things not completely clear to me. Specifically, it is not clear which antibody was used in figure 4 upper panels...was the L15 or L31? The one written is L15, and it seems specific, but in the previous figure was claimed as aspecific in some regions and in the results section it is written that for the rest of the study (figure 4 or 5 onwards?) it was used a different one (L31). It is also not clear to me why the NH one was discarded or of NH Ab and L31 are the same... Could the authors check these points and help the reader to correctly interpret this section of results and corresponding figures?

Answer:

As it is indicated in Figure 4 and the legend, upper panels show labeling with L15 antibody that provides selective CB1R labeling, but in parallel, may artifactually stain radial glia (Figures 2 and 3). It is also said in the text (Page4, lines 128-131): “…antibodies generated against last 15 amino acids (L15) of CB1R C-terminus also stain radial processes in addition to the previously demonstrated selective labeling of CB1R in cortico-fugal axons and interneuron cell bodies (Morozov et al., 2009; Keimpema et al., 2010).” Facilitating correct interpretation, we included the names of antibodies through the text and figure legends where possible.

We also included in Figure 4 new panels demonstrating control labeling with L31 antibody.

Round 2

Reviewer 1 Report

With the revised manuscript, my concerns are clarified.
I think there are enough information to be published.